# Disability and Participation in Breast and Cervical Cancer Screening: A Systematic Review and Meta-Analysis

**DOI:** 10.3390/ijerph19159465

**Published:** 2022-08-02

**Authors:** Fahrin Ramadan Andiwijaya, Calum Davey, Khaoula Bessame, Abdourahmane Ndong, Hannah Kuper

**Affiliations:** 1Department of Public Health and Preventive Medicine, Faculty of Medicine, University of Mataram, Mataram 83125, Indonesia; 2International Centre for Evidence in Disability, London School of Hygiene & Tropical Medicine, London WC1E 7HT, UK; calum.davey@lshtm.ac.uk (C.D.); hannah.kuper@lshtm.ac.uk (H.K.); 3West Hertfordshire NHS Trust, Watford WD18 0HB, UK; khaoula.bessame@nhs.net; 4Faculty of Health Sciences, Gaston Berger University of Saint-Louis, Saint-Louis 234, Senegal; abdourahmane.ndong@ugb.edu.sn

**Keywords:** disability, cancer, screening, mammography, pap smear

## Abstract

It is well established that access to preventative care, such as breast or cervical cancer screening, can reduce morbidity and mortality. Certain groups may be missed out of these healthcare services, such as women with disabilities, as they face many access barriers due to underlying inequalities and negative attitudes. However, the data have not been reviewed on whether women with disabilities face inequalities in the uptake of these services. A systematic review and meta-analysis were conducted to compare the uptake of breast and cervical cancer screening in women with and without disabilities. A search was conducted in July 2021 across four databases: PubMed, MEDLINE, Global Health, and CINAHL. Quantitative studies comparing the uptake of breast or cervical cancer screening between women with and without disabilities were eligible. Twenty-nine studies were included, all from high-income settings. One third of the 29 studies (34.5%, *n =* 10) were deemed to have a high risk of bias, and the remainder a low risk of bias. The pooled estimates showed that women with disabilities have 0.78 (95% CI: 0.72–0.84) lower odds of attending breast cancer screening and have 0.63 (95% CI: 0.45–0.88) lower odds of attending cervical cancer screening, compared to women without disabilities. In conclusion, women with disabilities face disparities in receipt of preventative cancer care. There is consequently an urgent need to evaluate and improve the inclusivity of cancer screening programs and thereby prevent avoidable morbidity and mortality.

## 1. Introduction

Breast and cervical cancer are leading causes of cancer death in women, accounting for 15.5% and 7.7% of all cancer deaths, respectively [1]. The early detection of breast or cervical cancer significantly improves the prognosis. Participation in a cancer screening programme is consequently associated with an 89% reduction in cervical cancer mortality [2] and a 21–25% reduction in breast cancer mortality [3]. However, there is strong evidence that disparities exist in cancer screening uptakes, even in settings where cancer screening programmes are well-established [4]. As a result, countries are failing to reach their cancer screening targets and people are dying unnecessarily [5].

Disability is potentially an important predictor of cancer screening uptake. People with disabilities face a range of barriers to accessing screening, including a lack of accessibility (information, transport, equipment, and facilities), a lack of affordability, communication difficulties, and negative attitudes of healthcare professionals [6,7,8]. They are also on average poorer and with less education, two known predictors of low screening uptake [4]. These barriers are likely to translate into lower service coverage and there is growing evidence suggesting that cancer screening uptake is lower among people with disabilities [6]. For instance, a study from the UK showed that women with disabilities are 36% less likely to attend breast screening and 25% less likely to attend bowel cancer screening, when compared to women without disabilities [9]. Women with multiple difficulties, or difficulties with vision or self-care were least likely to attend screening. Similarly, a national study in South Korea showed that having any type of disability is associated with 29% lower odds of cervical cancer screening [10].

The lower uptake of screening among people with disabilities is an important issue, as globally there are at least one billion persons with disabilities [11]. In the UK alone, there are at least 11 million people with disabilities [12]. Disability is particularly common in older people, who are also the focus of cancer screening programmes. However, there are no systematic reviews on the association between disability with breast and cervical cancer screening uptake, except one from 2013 that focused on women in the USA only [13]. The aim of this paper is to systematically review the global data on disparities in uptake of breast and cervical cancer screening among women in relation to their disability status.

## 2. Materials and Methods

This systematic review of peer-reviewed articles that presents research findings on the uptakes or receipt of breast or cervical cancer screening in women with and without disabilities was conducted in July 2021. This review used both the Preferred Reporting Items for Systematic Reviews and Meta-Analyses (PRISMA) and the Guidelines for Meta-Analyses and Systematic Reviews of Observational Studies (MOOSE Guideline) for conducting and reporting [14,15].

### 2.1. Defining Disability

This review used the World Health Organization International Classification Functioning, Disability, and Health (WHO-ICF) framework in defining disability [16]. Therefore, disability was classified as any form of physical, sensory, cognitive, or psychosocial impairment associated with activity limitations or participation restrictions. Moreover, this review also included diagnostic codes for specific illnesses (e.g., psychosis) or impairment (e.g., visual impairment, functional hearing loss) considered likely to be disabling.

### 2.2. Outcome

The primary outcome of interest was uptake/receipt of either cervical or breast cancer screening, comparing women with and without disabilities.

### 2.3. Data Sources and Search Terms

The literature search was conducted up to July 2021 across four databases (MEDLINE, PubMed, CINAHL, and Global Health). A combination of subject headings and key terms were used to assess: (1) disability, (2) breast or cervical cancer, and (3) screening uptake or utilization. A string of search terminologies was constructed, to ensure a comprehensive and holistic search strategy, such as the term “cancer screening”; where this can include early diagnostics or detection, also the type of diagnostic methods (Pap smear, mammography), where applicable truncation was also utilised. Boolean operators (‘AND’, ‘OR’, and ‘NOT’) were used to string search terminologies together. This review used both persons with disabilities and women with disabilities in the search term, even though breast and cervical cancer already implies women as the target population. The search strategy for MEDLINE is included as a Appendix A.

### 2.4. Inclusion Criteria

Studies were included if they quantitatively assessed the uptake and/or receipt of breast cancer screening (mammography, other radiological or laboratory examinations) or cervical cancer screening (Pap smear or visual inspection) in women with and without disability, aged 18–70. There were no limitations on study design (except the exclusion of qualitative studies); therefore, both observational and interventional studies were included. Eligible publication date was restricted to be between 2011 and 2021, to ensure relevance of the findings. 

We excluded studies if they were not written in English; not peer reviewed, or did not compare disparities of uptakes in women with disabilities to women without disabilities. Moreover, review studies and studies lacking clarity in reporting of measure of effect (i.e., no information on lower or upper limits, or ability to calculate these values), were also excluded from this review.

### 2.5. Study Selection

After the search strategy developed for MEDLINE was deemed to provide sufficient results, it was transferred to other search databases. Results from the database searches were transferred to Mendeley, which automatically removed duplicates. Subsequently, articles were transferred to Rayyan for title, abstract, and keyword screening. The initial screening was conducted by a single reviewer, and the results were checked by a second reviewer. The full texts were then retrieved and were screened by two authors (FRA and AN) according to the eligibility criteria for this review.

### 2.6. Data Extraction

There were three main components extracted from the selected articles: (1) article information (author information, country, and study design); (2) participant information (disability assessment, number of participants screened and did not receive screening, and contextual setting); (3) outcome (measurement and measure of association). Odds Ratios were extracted, rather than calculated from data presented, for this review.

### 2.7. Risk of Bias Assessment

Appraisal of risk of bias of included studies was undertaken using the Joanna Briggs Institute (JBI) Critical Appraisal Checklist for observational studies [17]. The checklist was used to examine methodological components of each study, and the studies were scored as to whether they had a low, medium, or high risk of bias.

### 2.8. Data Synthesis and Meta-Analysis

Estimates were pooled based on type of screening, resulting in the estimate of odds ratio of breast or cervical cancer screening comparing women with and without disabilities. The pooled estimates were calculated using a random effects model, as variations between included studies (e.g., country setting, sampling method, types of disability, and outcome measurement) can result in between-study heterogeneity. Heterogeneity across analyses was assessed using the *I*^2^ statistic.

Sub-group analyses were conducted for studies that comprised similar characteristics: type of disability and study methodology or design. Additionally, studies that were deemed with high risk of bias were excluded from the subgroup analyses. All statistical analyses were conducted using R version 1.4.1717 and package Meta [18].

## 3. Results

The database search was conducted on 21 July 2021 and resulted in 1037 papers (Figure 1). After duplicates were removed, 1037 titles were screened and ineligible studies were excluded. Consequently, 211 abstracts were screened, of which 35 full texts were selected for assessment. However, the full texts of two articles could not be retrieved. A further of six studies were excluded, due to incomplete information on the measure of effects or lack of reporting of differences between women with and without disabilities. Overall, 29 eligible studies were identified. No studies were added after conducting a further hand search and a reverse citation search.

### 3.1. Study Characteristics

Table 1 summarizes the characteristics of the 29 studies included in the review. Thirteen studies (45%) focused on breast cancer screening alone, six (21%) on cervical screening alone, and ten (34%) addressed both breast and cervical screening uptake. All included studies were conducted in high-income countries. The greatest number were conducted in the US (37.7% of all the studies), followed by the UK (13.8%), Canada, South Korea, and Northern Ireland (10.3% each). Australia, Denmark, and Sweden contributed one study each.

For studies that included cervical cancer screening, participants’ starting age was generally younger (starting from 18) compared to studies on breast cancer screening (minimum age of 40 and above).

#### 3.1.1. Study Design

Most of the included studies used a cohort study design (65.5%), utilizing data retrieved from national databases such as the National Health Insurance or Disability databases. Five included studies used a cross-sectional design (31%) [19,20,21,22,23] and one study used a mixed-method design [24].

#### 3.1.2. Types of Disabilities

The most common type of disability included in this review was psychosocial disability, accounting for 47% of all studies (*n =* 15), assessed as a psychiatric or mental health diagnosis, a history of psychiatric prescription, or self-reported mental status [24,25,26,27,28,29,30,31,32,33,34,35,36]. One third (31.3%, *n =* 10) of the studies included used disability in general or combining different types of disability into a category of having a disability or no disability. Few studies (9.7%, *n =* 3) focused on vision impairment, intellectual, or learning disabilities (9.7% *n =* 3), physical impairment (6.9%, *n =* 2), and functional hearing loss (6.9%, *n =* 2). Four studies also considered the number of disabilities and the severity of disability in relation with breast or cervical cancer screening.

#### 3.1.3. Outcome Measurement

The outcome of interest in this review is the uptake of breast or cervical cancer screening services. Of all the studies included in this review, 72% (*n =* 23) of the studies retrieved data on breast and cervical cancer screening utilization from central or national databases, where billing codes and examination or diagnostic history were linked to data on disability status. The other method of measurement was self-reporting through a questionnaire or interview (Table 2 and Table 3).

Three studies measured the adherence of women with disability to national cancer screening guidelines, thus, to be categorized as “screened” or “have utilized the service” if they had met criteria for the number of visits [26,35,45]. Table 2 and Table 3 describes how screening uptakes were measured in each study and its measure of association between disability status and breast or cervical cancer screening.

Generally, the included studies all presented ORs with a 95% confidence interval to estimate the measure of association between disability status and the utilization of breast or cervical screening services. Only two studies used the incidence rate ratio (IRR) to estimate the use of the cancer screening services comparing women with and without disabilities [25,32]. Some studies provided two estimates, depending on how disability status was coded, participants settings, and the severity of disability or impairment.

#### 3.1.4. Risk of Bias

One third of the 29 studies (34.5%, *n =* 10) were deemed to have a high risk of bias, and the remainder a low risk of bias (Table 2 and Table 3). Generally, studies that have a high risk of bias was due to not providing sufficient information on how the population or participants were categorized as exposed, and how the outcome was measured. Many studies were also marked down for using a self-reporting questionnaire as the sole measure of uptake or receipt of cancer screening.

#### 3.1.5. Breast Cancer Screening Uptake in Women with Disability

There were 28 data points included in the pooled analysis for breast cancer screening uptake, taken from 21 studies. The pooled estimate showed that women with disability have 0.78 (95% CI: 0.72–0.85) lower odds of breast cancer screening compared to women without disability. Individual estimates ranged from 0.49 to 1.22, and there was strong evidence for heterogeneity (*I*^2^ = 100%, *p* < 0.001) (Figure 2).

#### 3.1.6. Cervical Cancer Screening

There were 16 data points included in the pooled analysis for cervical cancer screening uptake, taken from 13 different studies (Figure 3). The overall pooled estimate of aORs is 0.67 (95% CI, 0.47–0.94), showing that women with disabilities have 0.67 lower odds of receiving cervical cancer screening compared to women without disability. From all the data points, only two data points showed significantly lower screening odds in women with disabilities. Moreover, one data point showed significantly higher screening in women with disabilities. There was evidence of high between-study heterogeneity (*I*^2^ = 100%, *p* ≤ 0.001).

### 3.2. Subgroup Analyses

Limited subgroup analyses were possible (Table 4).

For breast cancer screening, the pooled estimate of five data points from three studies showed that women with visual impairment had a 0.63 (95% CI: 0.51–0.77) lower odds of breast cancer screening compared to women without visual impairment. There was evidence of high between-study heterogeneity (*I*^2^ = 95%, *p* ≤ 0.001) (Figure 4). For psychosocial disabilities, the pooled estimate was an OR of 0.69 (95%CI, 0.60–80) (Figure 5), across seven data points from seven different studies.

For cervical cancer screening, the pooled estimate for the association of psychosocial disability and screening uptake was 0.64 (95%CI, 0.34–1.18) (Figure 6), using seven data points. Individual estimates ranged from 0.21 to 1.14. Furthermore, all sub-group analysis showed evidence of heterogeneity among each study.

## 4. Discussion

This systematic review and meta-analysis identified 29 studies across 8 different countries, evaluating the uptake of breast and cervical cancer screening by disability status. All studies included in this review were conducted in high-income countries. Overall, women with disabilities were 22% less likely to undergo breast cancer screening and 33% less likely to attend for cervical cancer screening compared to women without disabilities. The individual study results followed this pattern and out of the 29 studies only 3 did not show lower screening among women with disabilities.

The results of this review are consistent with the broader literature on this topic. A 2013 systematic review on cervical and breast screening and disability in the USA only identified five studies [13]. It showed evidence for a lower uptake of mammography among women with disabilities, but the evidence for clinical breast examination or cervical cancer screening was less clear. Qualitative data shows that women with disabilities report multiple barriers to accessing breast and cervical cancer screening, including physical barriers, cost, a lack of knowledge, fear, and attitudes of healthcare workers [46,47,48]. Studies have also shown that colorectal cancer screening is less frequent among people with disabilities compared to those without [49,50]. More broadly, it is well established that people with disabilities face greater challenges in accessing health care services [51].

This review showed clear evidence of disparities in breast and cervical cancer screening services experienced by women with disabilities. These findings further emphasize the importance of an inclusive cancer screening program and accessible healthcare services. However, this review did not provide details on the quality and effectiveness of healthcare received. Additionally, the review only found studies from high-income countries, making generalizability an issue. More evidence is therefore needed from low- and middle-income countries. The results do indicate that it is appropriate to design and evaluate interventions to improve cancer screening uptake among women with disabilities. These interventions should address the common barriers encountered, for instance, through providing training on disability to screening care providers, encouraging carers to support screening uptake and ensuring facilities and information are accessible. It may also be important to review policies and targets related to cancer screening to ensure that they are inclusive of people with disabilities. If these changes are not made, then women with disabilities will continue to have lower screening rates and face avoidable cancer-related deaths.

There are strengths and limitations of this review that should be taken into account when interpreting the results. The level of heterogeneity was high, likely because of differences in the measurement of disability between studies. Most studies used clinical diagnosis, such as vision impairment, hearing, and other psychiatric or mental diagnoses, and only one study explicitly explored disability through the ICF framework [17]. Another limitation is that most screening and data extraction was conducted by a single reviewer, which can potentially lead to selection bias. Additionally, this review only included studies published in English and did not explore the grey literature. In terms of strengths, the search strategy implemented a holistic approach to the definition of disability, which was achieved by using search terms that were in-line with the ICF framework. Furthermore, it included clinical diagnosis and conditions that reflect disability. Adhering to the PRISMA and MOOSE guideline also provided this review with methodological rigor.

## 5. Conclusions

Women with disabilities face disparities in receipt of preventative cancer care. There is consequently an urgent need to evaluate and improve the inclusivity of cancer screening programs and thereby prevent avoidable morbidity and mortality.

## Figures and Tables

**Figure 1 ijerph-19-09465-f001:**
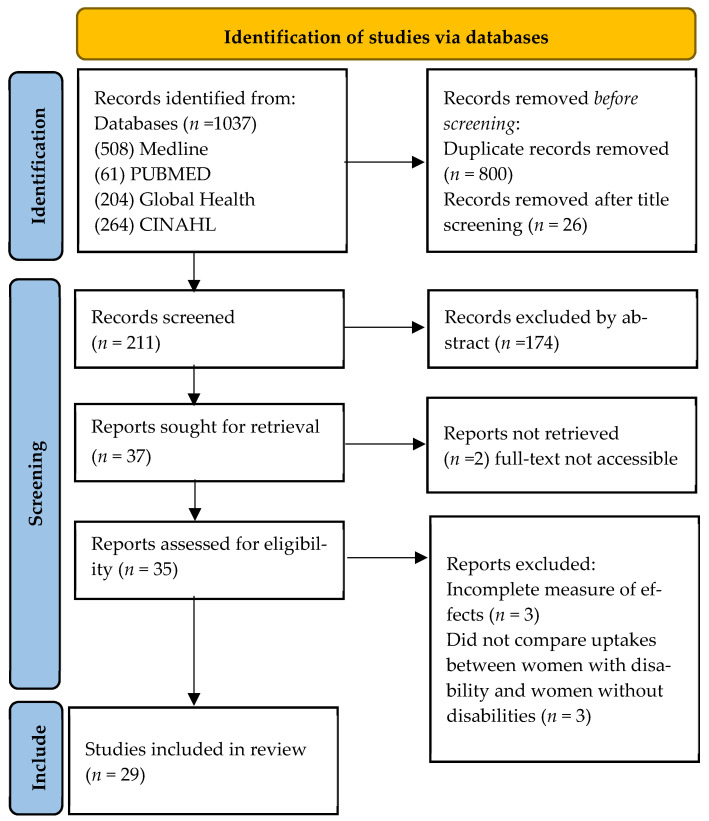
PRISMA flow chart of search results.

**Figure 2 ijerph-19-09465-f002:**
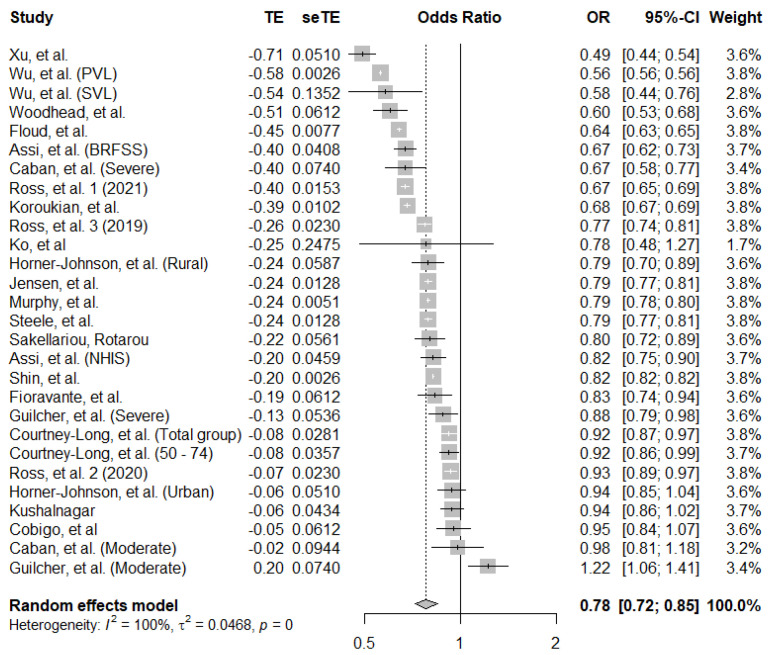
Pooled adjusted odds ratio estimates of breast cancer screening uptake by disability status [6,9,19,20,21,22,23,26,27,28,33,34,35,36,37,38,39,40,41,42,43,44].

**Figure 3 ijerph-19-09465-f003:**
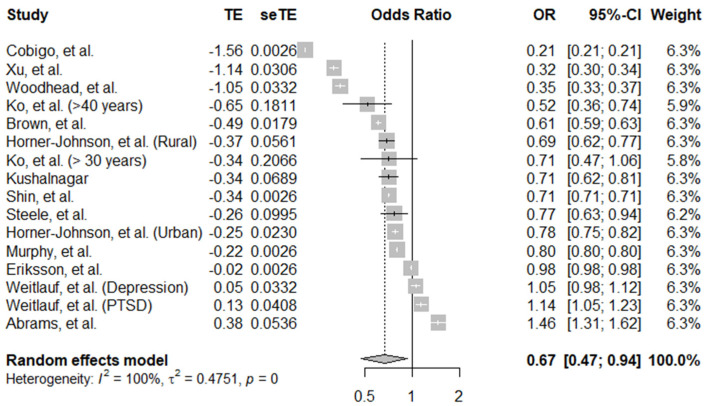
Pooled adjusted odds ratio estimates of cervical cancer screening uptake by disability status [19,22,24,26,29,30,31,34,36,37,38,39,43].

**Figure 4 ijerph-19-09465-f004:**
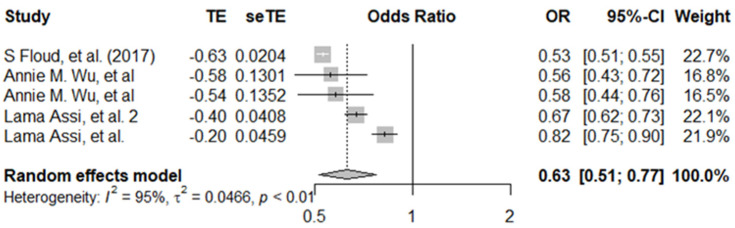
Pooled adjusted odds ratio estimates of breast cancer screening uptake by visual impairment status [9,23,44].

**Figure 5 ijerph-19-09465-f005:**
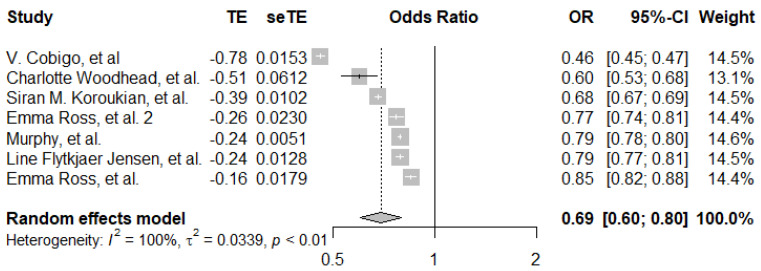
Pooled adjusted odds ratio estimates of breast cancer screening uptake by psychosocial disability status [26,27,28,33,34,35,36].

**Figure 6 ijerph-19-09465-f006:**
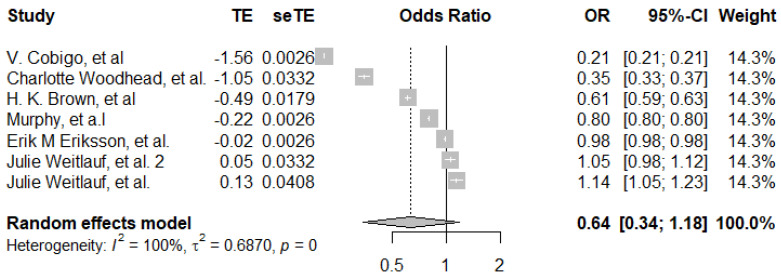
Pooled adjusted odds ratio estimates of cervical cancer screening uptake by psychosocial disability status [26,29,30,31,34,36].

**Table 1 ijerph-19-09465-t001:** Information and characteristics of included studies.

Author	Study Location	Study Design	Type of Disability	Definition of Disability	Type of Screening	Participants		Age Range (Years)	Follow up Time
						With Disability	Without Disability		
Cobigo et al. (2013) [36]	Canada	Cohort	Learning	Intellectual developmental disabilities based on the ICD-10.	Both	17,777	1,440,962	20–69	Breast: 2 years, Cervix: 3 years
Horner-Johnson et al. (2015) [37]	USA	Cohort	Functional	Presence of limitations in basic actions involving physical functions, vision, hearing, or cognition.	Both	10,985 (urban), 3108 (rural)	42,834 (urban), 8579 (rural)	Breast: 40–64, Cervix: 18–64	6 years
Ko et al. (2011) [22]	South Korea	Cross-sectional	Physical and psychosocial.	ICF: Physical, internal organ, and mental.	Both	23,511	11,660	42–69	2 years
Kushalnagar. (2019) [38]	USA	Cross-sectional	Hearing	Functional hearing impairment.	Both	Breast: 324 Cervix: 529	Breast: 1086 Cervix: 1119	Breast: 40–74 Cervix: 21–65	n/a
Murphy et al. (2021) [34]	USA	Mixed methods, retrospective for quantitative; and qualitative.	Psychosocial	Serious mental illness (SMI): schizophrenia, bipolar depression, major depression.	Both	Breast 94,921 Cervix 274,643	Breast: 11,955,674 Cervix: 31,949,537	21–64	7 years
Osborn et al. (2012) [25]	UK	Cohort	Learning	General terms and related terms (e.g., autism, down syndrome, and Fragile X syndrome).	Both	Breast: 2956 Cervix: 6254	50,779	Breast: 50–64 Cervix: 20–65	10 years
Steele et al. (2017) [19]	USA	Cross-sectional	Physical and functional.	Self-report of disability.	Both	2580	12,499	21–75	n/a
Woodhead et al. (2016) [26]	UK	Cross-sectional	Psychosocial	Serious mental illness based on ICD-10 diagnosis.	Both	Breast: 625, Cervix: 1393	Breast: 25,385, Cervix: 106,554	Breast: 50–70, Cervix: 25–64	n/a
Xu et al. (2017) [39]	USA	Cohort	Visual	Clinical diagnosis of visual impairment.	Both	Breast: 1308, Cervix: 1247	Breast: 2635, Cervix: 2483	Breast: 40–74, Cervix: 20–74	11 years
Assi et al. (2020) [23]	USA	Cross-sectional	Visual	Self-reported visual impairments.	Breast	1915	10,205	50–74	n/a
Caban et al. (2011) [40]	USA	Cohort	Functional and psychosocial.	Reported functional limitations of activity of daily living (ADL) and instrumental activities of daily living (IADL).	Breast	2281	2329	>65	2 years
Courtney-Long et al. (2011) [41]	USA	Cross-sectional	Physical and functional.	Self-report of disability.	Breast	64,905	130,394	40–74	2 years
Fioravante et al. (2021) [21]	USA	Cross-sectional	Hearing	Functional hearing loss.	Breast	2123	10,067	50–74	n/a
Floud et al. (2017) [9]	UK	Cohort	Functional (including psychological) and physical.	Self-report of disability.	Breast	109,869	363,316	50–70	5 years
Guilcher et al. (2014) [42]	Canada	Cohort	Physical and functional.	Morbidity: presence of limiting disease, e.g., arthritis, hypertension.	Breast	4660	5703	50–69	2 years
Jensen et al. (2016) [33]	Denmark	Cohort	Psychosocial	Schizophrenia, affective disorders, eating disorder.	Breast	47,648	96,616	50–69	Up to 10 years
Ross et al. (2020) [6]	Northern Ireland	Cohort	Physical and psychosocial.	Self-report of disability.	Breast	20,541	36,787	48–70	1 year
Ross et al. (2020) [28]	Northern Ireland	Cohort	Psychosocial	Chronic poor mental health.	Breast	6162	51,166	50–70	4 years
Ross et al. (2021) [27]	Northern Ireland	Cohort	Psychosocial	Record of psychotropic prescription.	Breast	17,521	39,807	50–70	3 years
Sakellariou and Rotarou. (2019) [20]	UK	Cross-sectional	Physical	Lower limb impairment.	Breast	2697	6794	20–70+	n/a
Shin et al. (2020) [43]	South Korea	Cohort	Physical and psychosocial.	Diagnosis of disability by healthcare professional.	Breast	419,376	5,864,247	>40	10 years
Koroukian et al. (2012) [35]	USA	Cohort	Psychosocial	Morbidity: presence of limiting disease, e.g., arthritis, hypertension.	Breast	61,661	68,427	50–64	n/a
Wu et al. (2021) [44]	USA	Cohort	Visual	Partial vision loss (PVL) and severe vision loss (SVL).	Breast	PVL: 348, SVL: 348	348	65–72	5 years
Abrams et al. (2012) [24]	USA	Cohort	Psychosocial	Psychosis (schizophrenia), substance use disorder, bipolar disorder, or mania.	Cervical	20,306	85,375	19–64	1 year
Brown et al. (2016) [30]	Canada	Cohort	Intellectual and developmental.	Clinical diagnosis of intellectual and developmental disabilities.	Cervical	5033	527,437	20–64	n/a
Eriksson et al. (2019) [29]	Sweden	Cohort	Psychosocial	Psychiatric diagnosis.	Cervical	65,292	341,171	23–60	5 years
Shin et al. (2018) [10]	South Korea	Cohort	Physical and functional.	Diagnosis of disability by healthcare professional.	Cervical	426,189	7,376,529	>50	10 years
Tuesley et al. (2019) [32]	Australia	Cohort	Psychosocial	Classified as serious mental illnesses, based on prescriptions in the last 12 months.	Cervical	18,363	899,777	18–69	10 years
Weitlauf et al. (2013) [31]	USA	Cohort	Psychosocial	PTSD and depression based on clinical diagnosis ICD 9.	Cervical	17,295	16,828	18–65	1 year

Breast: breast cancer screening. Cervix: cervical cancer screening.

**Table 2 ijerph-19-09465-t002:** Details on outcome measurement and estimate of association between presence of any disability in women and breast cancer screening.

Author	Outcome Definition	Assessment Method	Uptake (%)		Unadjusted OR	aOR (95% CI)	Risk of Bias Rating
			Women with Disabilities	Women without Disabilities			
Assi et al. [23]	Receipt of mammography in the past 2 years.	Self-report	(−5.02%) difference in proportions		BRFSS: 0.63 (0.56–0.70)	0.67 (0.59–0.75)	High
					NHIS: 0.78 (0.68–0.89)	0.82 (0.71–0.89)	
Caban et al. [40]	Receipt of mammography in the past 1 year of the study period.	Self-report	n/a	n/a	Moderate disability: 0.76 (0.64–0.91)	0.98 (0.81–1.18)	Low
					Severe disability: 0.46 (0.40–0.54)	0.67 (0.54–0.83)	
Cobigo et al. [36]	Receipt of mammography in the past 2 years.	Clinical record (Insurance code)	42%	60%	0.47 (0.45–0.50)	0.95 (0.84–1.08)	Low
Courtney-Long et al. [41]	Receipt of mammogram within the past 2 years.	Self-report	Total group: 72%	78%	n/a	0.92 (0.87–0.98)	High
			Aged 50–74: 78%	83%	n/a	0.92 (0.85–0.99)	
Fioravante et al. [21]	Receipt of mammogram within past two years.	Self-report	n/a	n/a	0.84 (0.73–0.96)	0.83 (0.72–0.96)	High
Floud et al. [9]	Clinical registration of breast cancer screening in the past 3 years.	Clinical record	83%	89%	n/a	0.64 (0.62–0.65)	Low
Guilcher, et al. [42]	Receipt of mammography within two years.	Clinical record	Moderate disability: 67%	68%	n/a	1.22 (1.09–1.38)	Low
			Severe disability: 67%	68%	n/a	0.88 (0.78–0.99)	
Horner-Johnson et al. [37]	Receipt of mammography within two years.	Clinical record	Rural: 67%	70%	0.63 (0.56–0.72)	0.79 (0.68–0.91)	Low
			Urban: 73%	76%	0.85 (0.77–0.93)	0.94 (0.84–1.04)	
Jensen et al. [33]	Rates of participation in the first 18 months of the screening round.	Clinical record	74.5%	81%	0.65 (0.63–0.68)	0.79 (0.77–0.82)	Low
Ko et al. [22]	Utilisation of breast cancer screening services during the study period.	Self-report	26%	32%	n/a	0.78 (0.43–1.4)	
Koroukian et al. [35]	Receipt of screening mammography in the study period and adherence to national guideline.	Clinical record	38%	32%	n/a	0.68 (0.66–0.7)	Low
Kushalnagar [38]	Adherence to mammography guidelines.	Self-report	76%	82%	n/a	0.94 (0.77–0.94)	Low
Murphy et al. [34]	Receipt of breast cancer screening, during the 6 year study period.	Clinical record	51%	62%	0.88 (0.87–0.89)	0.79 (0.78–0.8)	Low
Osborn et al. [25]	Clinical record of attending for mammography or mammography results during the study period.	Clinical record	44%	52%	IRR = 0.78 (0.74–0.83) *	IRR = 0.76 (0.72–0.81) *	Low
Ross et al. (2021) [27]	Records of women attending the screening programme from 1 April 2011 to 31 March 2014.	Clinical record	74%	81%	0.71 (0.68–0.74)	0.67 (0.64–0.7)	Low
Ross et al. (2020) [28]	Clinical attendance of screening invitation.	Clinical record	75%	81%	0.53 (0.50–0.57)	0.93 (0.89–0.98)	Low
Ross et al. (2019) [6]	Breast cancer screening attendance.	Self-report	68%	80%	0.67 (0.64–0.70)	0.77 (0.73–0.82)	Low
Sakellariou, Rotarou [20]	Receipt of mammogram within the past three years.	Secondary data analysis	48%	46%	n/a	0.80 (0.70–0.92)	
Shin et al. [43]	Clinical attendance or use of mammography for breast cancer screening during 2014–2015.	Clinical record	41%	54%	n/a	0.82 (0.82–0.83)	Low
Steele et al. [19]	Receipt of mammogram within the past 2 years.	Self-report	67%	73%	n/a	0.79 (0.77–0.82)	High
Woodhead et al. [26]	Receipt of mammography in the past three years, and for aged 50–64 in five years.	Clinical record	58%	66%	0.72 (0.61–0.86)	0.60 (0.49–0.73)	Low
Wu et al. [44]	Receipt of mammogram within past two years.	Insurance record	PVL: 77%	81%	n/a	0.56 (0.36–0.87)	Low
			SVL: 72%	81%	n/a	0.58 (0.37–0.9)	
Xu et al. [39]	Full adherence or partial adherence to screening guidelines, during the study period.	Insurance record	65%	75%	n/a	0.49 (0.40–0.6)	Low

aOR: adjusted Odds Ratio. OR: Odds ratio. n/a: Not available. *: reported as IRR.

**Table 3 ijerph-19-09465-t003:** Details on outcome measurement and estimate of association between presence of any type of disability in women and cervical cancer screening.

Author	Outcome Definition	Assessment Methods	Uptake (%)				Risk of Bias Rating
			Women with Disabilities	Women without Disabilities	Unadjusted OR (95% CI)	aOR (95%CI)	
Abrams et al. [24]	Clinical attendance to cervical screening over the study period (July 2004–June 2004).	Clinical record	25%	18%	n/a	1.46 (1.36–1.57)	Low
Brown et al. [30]	Clinical attendance to cervical screening between 1 April 2007 and 31 March 2010.	Clinical Record (Insurance code)	68%	77%	n/a	0.61 (0.58–0.65)	Low
Cobigo et al. [36]	Receipt of at least one Pap test over a 3 year period.	Clinical record	34%	67%	0.26 (0.25–0.27)	0.21 (0.2–0.21)	Low
Eriksson et al. [29]	Clinical participation in cervical cancer screening over the 5 year study cohort period.	Clinical record	86%	89%	n/a	0.98 (0.97–0.98)	Low
Horner-Johnson et al. [37]	Receipt of Pap smear with three years.	Clinical record	Rural: 77%	84%	0.50 (0.44–0.58)	0.69 (0.59–0.81)	Low
			Urban: 82%	87%	0.67 (0.62–0.72)	0.78 (0.87–0.96)	
Ko et al. [22]	Utilisation of cervical cancer screening services during the study period.	Self-report	>30 years: 29%	45%	n/a	0.71 (0.41–1.22)	High
			>40 years: 23%	43%	n/a	0.52 (0.27–0.98)	
Kushalnagar [38]	Adherence to pap smear guidelines.	Self-report	78%	85%	n/a	0.71 (0.59–0.86)	High
Murphy et al. [34]	Receipt of pap smear during the 6 year study period.	Clinical record	52%	61%	0.92 (0.92–0.93)	0.80 (0.80–0.81)	Low
Osborn et al. [25]	Clinical record of attending for Pap smear during the study period.	Clinical record	68%	85%	IRR = 0.55 (0.53–0.57) *	IRR = 0.54 (0.52–0.56) *	Low
Shin et al. [43]	Use of the cervical cancer screening programme in the past ten years (2006–2015).	Administrative data (clinical record)	54%	60%	n/a	0.71 (0.71–0.72)	Low
Steele et al. [19]	Receipt of a Pap smear within the past 2 years.	Self-report	72%	82%	n/a	0.77 (0.60–0.99)	High
Weitlauf et al. [31]	Use of Pap smear test in outpatient setting during the study period.	Insurance record	n/a		Depression: 1.04 (0.98–1.09)	1.05 (0.99–1.12)	Low
					PTSD: 1.17 (1.09–1.26)	1.14 (1.06–1.22)	
Woodhead et al. [26]	Receipt of cervical cancer screening any time in the last three years for those aged up to 49 years, or any time in the last five years for those aged 50–64.	Clinical record	80%	78%	1.16 (0.99–1.35)	0.35 (0.29–0.42)	Low
Xu et al. [39]	Full adherence or partial adherence to screening guidelines during the study period.	Insurance record	64%	81%	n/a	0.32 (0.27–0.39)	Low

aOR: adjusted Odds Ratio. OR: Odds ratio. n/a: not available. *: reported as IRR.

**Table 4 ijerph-19-09465-t004:** Summary of subgroup analyses.

Screening Type	Sub-Group	Studies Included (References)	Pooled Estimate (95% CI)	Heterogeneity (*I*^2^)
Breast cancer	Visual impairment	N = 3 studies [5,20,33]	0.63 (0.51–0.77)	95%
Breast cancer	Psychosocial	N = 7 studies [22,23,24,25,26,31,32]	0.69 (0.60–0.80)	100%
Cervical cancer	Psychosocial	N = 6 studies [22,24,27,28,29,32]	0.64 (0.34–1.18)	100%

## Data Availability

Data is available upon reasonable request.

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
