# Peer review of "Disability and Participation in Breast and Cervical Cancer Screening: A Systematic Review and Meta-Analysis"

_ijerph, 2022, doi:10.3390/ijerph19159465_

Round 1
Reviewer 1 Report
Dear authors,
This review showed clear evidence of disparities in breast and cervical cancer screening services experienced by women with disabilities. These findings further emphasise the importance of an inclusive cancer screening programme and accessible healthcare services.
I think your research is important for the future of clinicians and clinical researchers who aim to better understand how women with disabilities face many barriers when accessing healthcare services.
I was also impressed with how well the study was researched and analyzed. This manuscript was very helpful for me.
However, I have some questions below.
1. Please check the blank spaces in table 1 and table 2.
2. There were no descriptions of Figures S1, S2, and S3.
3. In the section of 3.3. subgroup analyses.
As you are analyzing the supplementary figures in your results, please include these figures in the main text (Supplement figures S1, S2, and S3).
4. Unify the results in the Abstract and Conclusion. Also, please add more detail to the Conclusion.
5. I also recommend you simplify the title.
For example, “A Systematic Review of Breast and Cervical Cancer Screening Uptake Lower among Women with Disabilities by Meta Analysis”
6. Finally, I recommend editing the English in your manuscript again.
For example, I don't really understand lines 16-19 in the abstract section.
Author Response
Please the attachment

Reviewer 2 Report
Thank you for the opportunity to review this manuscript. The study looked at the literature on breast and cervical cancer screening uptake between women with and without disabilities. The study findings highlight disparities in screening rates and the need to improve access. The paper is well written and appropriately analysed and reported. I have only minor comments.
Abstract
Page 1 line 17 “has not been reviewed whether…” change to “has not been reviewed on whether…”
Page 1 line 19: “A search”
Page 1 line 21: uptakes change to uptake
Introduction
Page 1 line 31: “Breast and cervical cancer are the leading causes of cancer death in women, contributing up to 15.5% and 7.7% of all cancer deaths, respectively[1].” Consider rewording as this is slightly misleading. Cervical cancer is the 4th leading cause of cancer death in women behind breast, lung and colorectal. But together they are the leading cause.
Material and methods
Page 2 section 2.1: Please include references for the definition of disability used. I am a little confused why the medical model of disability is used in the definition. Did the search include specific disabilities? If so how were these selected over others -from the text it seems that maybe visual impairment and psychosis were included but not Down Syndrome or Autism for example.
Page 2 section 2.3: Include search strategy for at least one of the databases in supplementary file.
Page 3 section 2.5: Since only a single reviewer was used to conduct the initial screen, did they consult on any or have a certain number checked by another reviewer?
Page 3 section 2.8: For Tables 2 and 3 were the ORs reported from the studies themselves or calculated for the present study?
Results
Page 4 line 138: should it say “after duplicates were removed, 237 titles were screened” rather than 800? There were 800 duplicates.
Page 5 line 188: change to “most common type of disability”
Page 5 section 3.1.2: Consider including intellectual disabilities here there were a few studies included with this category.
Table1: Include the references next to Author name eg Osborn et al. [25] so can easily identify in text.
Table 1: ADL is not defined.
Table 2: PVL has already been defined in Table 1.
Table 2 and 3: It is not clear what the terms in () next to author refer to eg Caban, et al. (moderate) as both moderate and severe are reported.
Page 4 line 237: reword “and/or outcome measurement” as is not clear what this refers to.
Page 4 line 240: change to “uptake”
Page 10 line 218: You mention that 2 studies used IRR rather than OR but in Table 2 and 3 OR is reported for [25]. Why was the outcome for [32] and 2 other cervical papers not reported (Table 3 has 13 papers extracted but should have 16)?
Page 11 line 243: From figure 2 the pooled estimate is 0.78 (95%CI: 0.72; 0.84) not 0.77 as reported in text.
Figure 2: Include qualifier for Courtney-Long et al. eg Courtney-Long et al. (Aged 50-74)
Page 13 line 315: remove “problem”
Reviewer 3 Report
This is a well-written and organized manuscript outlining a very important topic in breast and cervical cancer research. I think the authors did a really good job describing the strategy, methods, and results. I only have a few suggestions: 1) make better quality figures 2 and 3 (better resolution); 2) include a little bit more information on potential barriers to lower screening rates, such as Breast Cancer Screening Barriers and Disability (nih.gov)
Round 2
Reviewer 1 Report
Dear authors,
Thank you for submitting your revised manuscript. The authors revised the sentences according to my comments.
Therefore, I have no further comments to make, all of my previous concerns were adequately addressed. This manuscript will be satiating the reader's interest.